# Molecular Dynamics Simulations of Ion Transport through Protein Nanochannels in Peritoneal Dialysis

**DOI:** 10.3390/ijms241210074

**Published:** 2023-06-13

**Authors:** Jie Liu, Tao Zhang, Shuyu Sun

**Affiliations:** Computational Transport Phenomena Laboratory, Physical Science and Engineering Division (PSE), King Abdullah University of Science and Technology, Thuwal 23955-6900, Saudi Arabia; jie.liu.1@kaust.edu.sa

**Keywords:** ion transport, molecular dynamics, MDMC algorithm, protein channel

## Abstract

In recent decades, the development of dialysis techniques has greatly improved the survival rate of renal failure patients, and peritoneal dialysis is gradually showing dominance over hemodialysis. This method relies on the abundant membrane proteins in the peritoneum, avoiding the use of artificial semipermeable membranes, and the ion fluid transport is partly controlled by the protein nanochannels. Hence, this study investigated ion transport in these nanochannels by using molecular dynamics (MD) simulations and an MD Monte Carlo (MDMC) algorithm for a generalized protein nanochannel model and a saline fluid environment. The spatial distribution of ions was determined via MD simulations, and it agreed with that modeled via the MDMC method; the effects of simulation duration and external electronic fields were also explored to validate the MDMC algorithm. The specific atomic sequence within a nanochannel was visualized, which was the rare transport state during the ion transport process. The residence time was assessed through both methods to represent the involved dynamic process, and its values showed the temporal sequential order of different components in the nanochannel as follows: H_2_O > Na^+^ > Cl^−^. The accurate prediction using the MDMC method of the spatial and temporal properties proves its suitability to handle ion transport problems in protein nanochannels.

## 1. Introduction

Dialysis is a medical treatment that removes waste products and extra water from the body in the event of kidney failure [1,2]. In the last few years, almost half of elderly people in the United States have received dialysis [3]. Coronavirus disease 2019 patients also have the symptoms of kidney diseases [4], which means some patients with kidney failure also need dialysis treatment. Thus, more studies about dialysis are necessary and urgent.

Hemodialysis and peritoneal dialysis are common dialysis techniques, and the latter is more convenient for patients since it uses the peritoneum rather than an artificial semipermeable membrane [5,6,7]. Boehm et al. [8] proposed the first description of the peritoneal surface proteome during chronic peritoneal dialysis. Regarding the transport of water and water-soluble materials, Rippe et al. [9] found that the principal peritoneal exchange route was represented by protein-restrictive pores with a radius of several angstroms, and these pores accounted for ~99% of the pore area. The peritoneum has biological activity, and its membrane proteins play an important role in it [10]. There are two ways of water transport across the peritoneum: lipid bilayers and protein channels [11]. The membrane proteins also control the passage of ions in the body fluid, and the protein channels are at the nanometer level [12]; therefore, conventional experimental methods are not suitable for studying transport in such protein nanochannels, although this is the common transport process in peritoneal dialysis. Thus, investigating the ion transport phenomenon at the nanoscale level is significant for the understanding of peritoneal dialysis.

MD simulations, which have already been utilized in different fields [13,14,15], are good at nanoscale calculations, and the atomic behavior can be simulated and shown as a molecular trajectory [16]. Lindahl et al. [17] studied membrane protein issues via MD simulations, and the behavior of proteins in lipid bilayers was visualized. The results showed that membrane proteins always perform with high freedom, and the protein nanochannels were hard to determine unless some parts of the protein molecules were constrained. Membrane nanochannel transport is controlled by different proteins, which usually means different transport mechanisms, such as voltage-gated channels, ligand-gated channels, mechanosensitive channels, outer membrane proteins, and transporters [18]; thus, an accurate full-atom interactive calculation during membrane transport is difficult. Hilder et al. [19] constructed a nanotube to mimic a protein nanochannel and understand its selective function, addressing the water transport problem [20]. In such an ideal molecular model, quantitative analysis is easier. Corry [21] studied the membrane transport of ions and water by using carbon nanotubes, and the results showed that water passed through tunnels with weak impediments. Hou et al. [22] reported that the oxygen atoms on the nanochannels provided nonbridging sites for the attraction of Na ions, which could form anion–cation pairs with Cl ions. Li et al. [23] investigated the transport of saline water through graphene nanochannels and examined the effect of external electric fields on the ionic Coulomb blockade. Therefore, the simplified nanochannel model is a good way to explore and validate the basic mechanism of membrane transport.

MD simulations can provide accurate results at the nanoscale; however, this is a cost-intensive method, the computational speed is limited by the system size [24,25], and it is the short slab for many particle methods in the Lagrangian frame [26,27,28,29,30]. The molecular weight of protein molecular systems ranges from thousands to millions, thus requiring further studies in larger systems with longer simulation times [31]. We proposed an MDMC algorithm in a previous work [32], demonstrating its suitability for accelerating molecular simulations. It utilizes the results produced via MD simulations as the ground truth and converts the MD trajectories into a probability transition matrix, which is the basis for the following Monte Carlo calculation [33]. The MDMC algorithm is also able to make coarsening operations from a spatial scale and temporal scale. The successful application of this MDMC algorithm could prove it to be an effective and novel tool for investigating ion transport within membrane protein nanochannels.

In this work, ion transport in protein nanochannels was studied by using MD and MDMC methods. The spatial distribution of ions within the protein nanochannels was calculated with the MDMC algorithm. The rare events during the passage of different ion components were also predicted. The paper is organized as follows. In Section 2, the nanochannel model and simulation methods are introduced. Section 3 describes the MD simulation results regarding the spatial behavior of ions and their prediction using the MDMC algorithm; the residence time is calculated to describe the dynamics property of the ion transport process. The conclusions are summarized in Section 4.

## 2. Results

### 2.1. Spatial Distributions of Ions in the Protein Nanochannel

Before the ion transport simulation, the molecular model needed to be built in the computational domain. Ions are driven by an electric field force, and their moving state changes accordingly, which is a common phenomenon, whether in the human body [34,35] or the natural environment [36,37]. The spatial property plays a critical role to show the result. Here, the number density profiles of total fluid, H_2_O, Na ions, and Cl ions were collected during the MD simulation (Figure 1). With a mesh size of 0.5 Å, the simulation domain was chunked, and the density value was derived by counting the atomic numbers present in each chunk within the simulation domain. Due to van der Waals interactions between carbon and total fluid atoms, density peaks appeared at the interface between the fluid and graphene layers, and consequently, multilayers formed, as proved in previous studies [38,39]. It can be observed that the sodium ions are concentrated on the left side of the electrode due to the electric field force. However, the one-dimensional density profiles illustrated in Figure 1 did not show the spatial characteristics within the nanochannel. It will lose some details when the information is collected in the reduced dimension, especially for the information within the channel. Consequently, a two-dimensional contour of the number density was also derived (Figure 2). Furthermore, the fluid molecules within the nanochannel space exhibited a heterogeneous state as well as periodic occurrences that were caused by the stratified structure of the nanochannel as well as the adsorption layers. This structure also has a confined effect on the fluid molecules, especially at the beginning and ending positions of the nanochannel, which is similar to the phenomenon of ion rejection.

The most abundant fluid component is water, which is also the most likely to affect the distribution property of the fluid phase.

In order to investigate the energy mechanism of water in the nanochannel, the potential energy of water was calculated, as shown in Figure 3, and its potential energy layers corresponded to the multilayer layout displayed in Figure 2. At the beginning of the nanochannel, two potential energy wells were observed clearly; this phenomenon will be explained in the Discussion. Due to the driving force from the left side to the right, the water molecules in the nanochannel always collided with the right-side wall, leading to high potential energy. On the right side of the nanochannel, when a water molecule was passing through the nanochannel, there was also a potential energy well since this molecule had to overcome the resistance produced by the fluid on the right side. The ions will always be trapped in the initial part of the nanochannel. In addition, the heterogeneous potential energy distribution illustrates that the ion transport was inhomogeneous, suggesting that the ion rejection phenomenon can be presented via this molecular model.

### 2.2. Spatial State Predictions via the MDMC Algorithm in the Protein Nanochannel

The spatial distribution of Na^+^ was determined through MD simulations, as displayed in Figure 4. The atomic number was counted on each point as long as the atom appeared at the corresponding position; thus, the resulting value could be considered as the accumulated value. In the protein nanochannel molecular system, which always contains millions of atoms, this kind of simulation will require several computational resources. Here, an approximated model was adopted to save the simulation cost and facilitate the validation assessment. The prediction by the MDMC algorithm agreed well with the MD simulations; the variation profiles were well described, especially on the nanochannel boundaries.

The accumulated molecular number reveals the probability that one atom exists at relevant spatial points. That is, the higher its value, the higher the probability and state frequency in the MDMC calculation process. In Figure 5a,b, the two-dimensional number density contour of Na^+^ was also calculated and verified. The adsorption layers on the left and right sides matched. The random movement and the high-density parts in the nanochannel region were reproduced accurately, which cannot be visualized in one-dimensional profiles. On the left side, a striped state distribution was observed due to the Na^+^ movement since the moving process was fast compared to the total simulation time and the trajectories were collected in a striped shape. Figure 5c illustrates the differences between the MD and MDMC results. The successful match between these two methods demonstrates the feasibility of the MDMC algorithm for predicting the spatial properties of ion transport. However, two sides of the error in Figure 5c had different performances. The error performance in the left side was not as good as the result in the right side. In the right side, according to the MD results, the data distribution was more homogeneous, providing a better sampling dataset, while in the left side, the ions always transported from the left boundary to the channel wall, forming the stripe shape trajectory. Hence, the ions could not provide an ergodic trajectory, which means the dataset was not enough in the left side. Therefore, fewer data induce more errors, and the errors in Figure 5c occur.

## 3. Discussion

By using the MDMC method, it is possible to describe the spatial distribution of ions well, but ion transport is a dynamic problem, which poses a challenge to the MDMC method. The MC method is good at handling equilibrium problems, while ion transport is a nonequilibrium process and is usually described by using the MD method. During the MD simulation, the residence times of different components within the nanochannel differ due to the different electric field forces, and this characteristic provides the feasibility of describing the dynamics behavior with the MDMC algorithm. Following this idea, MD simulations with different durations were computed, and the corresponding residence times were also counted. Figure 6 compares the results of the MD and MDMC methods, showing a good agreement for the Na^+^ residence time, which confirms the applicability of the MDMC algorithm to this dynamics process as long as the dynamics property can be well described by using the relevant equilibrium state parameters.

Considering that ions are affected by electric field forces, it was also necessary to examine the electric field sensitivity of the ions. In order to investigate the area shown in the molecular model, electric field gradients ranging from 0.1 to 0.9 V·Å^−1^ were applied to the region. It can be seen from the results in Figure 7 that there was strong discontinuity. Under a low electric field gradient (0.1–0.3 V·Å^−1^), the ions did not obtain enough electric force to break through the nanochannel and, thus, the number of passing channel events was small, leading to a very short residence time. At higher electric field gradients (0.5–0.9 V·Å^−1^), the residence time declined, because of the stronger electrical driving force. At the electric field gradients of 0.5 V·Å^−1^, the result shows the highest residence time, because the ions were able to go through the nanochannel with these electric field gradients, but they could stay for a longer time within the nanochannel compared with a higher electric field gradient.

In the transport process, besides the event of one Na^+^ passing through the nanochannel alone, a kind of rare event also occurs. As shown in Figure 8, different components would follow a specific order to pass through the nanochannel; the sequence is H_2_O, Na^+^, and Cl^−^, from right to left. First, the nanochannel edges are modified by hydrogen atoms, which can form hydrogen bond interaction with the oxygen atoms of water. In this way, the water molecules are easily adsorbed at the nanochannel entrance and trapped after the first layer [21], as shown in Figure 8a. This trapped water molecule cannot move forward due to the lack of a driving force, but its oxygen atom would form Coulomb interactions with Na^+^, making this ion move to the nanochannel entrance, and this process is identified as Na^+^ relocation affected by the trapped water molecule [22]. Next, driven by the electric force, the Na^+^ is pushed into the nanochannel, and the trapped water molecule also moves into the next position. By following the same way, Na^+^ attracts the Cl^−^ on the left side, and, hence, Cl^−^ is also relocated at the nanochannel entrance. When the Na^+^ is driven forward, the trapped water molecule is pushed forward, while the Cl^−^ is dragged forward. The direct description of this dynamic behavior is difficult, but we know that the passing order differs for these three components, and the residence time should be reduced for all of them. Therefore, the corresponding residence time was calculated, as shown in Figure 8b, showing that the water had the longest residence time and Cl^−^ the shortest one, which agrees with the MD simulation results.

## 4. Materials and Methods

### 4.1. Molecular Model

In the constructed molecular model, as depicted in Figure 9, six graphene layers were set in its center to mimic an organic protein nanochannel. Their spacing distance was set at 3.4 Å to avoid nonphysical interlayer diffusion, and the nanochannel diameter was 5.405 Å [23]. Although the realistic protein model can provide more accurate results, the complication of protein molecules makes the model problem-dependent. Different polar atoms in proteins are able to induce various effects on the ion transport, which also brings a tricky result for the comparison. Thus, it is better to use a more general model to describe the basic transport process in order to verify the algorithm’s applicability. Graphene is good at being used to build a regular model, suggesting that the quantitative analysis is easy to address, which can provide a clearer comparison. The carbon atoms at the nanochannel edges were modified by using hydrogen atoms. The spatial size of the simulation domain was 26.1766 Å × 30.3968 Å × 68.0527 Å, and the periodic boundary conditions were applied in all directions. To make ion transport accessible, a constant electric field on Na^+^ was set in a certain part of the computational domain, as shown in Figure 9. Due to the periodic boundary conditions, the pressure of continuous fluid molecules was kept balanced on two sides. The model was visualized via the visual MD package [40].

### 4.2. MD Simulations

A polymer-consistent force field was applied on all atoms [41], and its parameters are listed in Table 1. The van der Waals interactions between different atoms were described using the Lennard–Jones equation [42], the parameters of different atom types were calculated via the Waldman–Hagler combining rules [43], and the electrostatic force was assessed with the Ewald method [44]. The cutoff distance was 1.2 nm in all simulations.

The large-scale atomic/molecular massively parallel simulator program (Mar 2020) was used for the MD simulations [45]. The graphene nanochannel was controlled as a rigid part, which maintained the system stability, and the hydrogen atoms modified on its edges were thermally controlled to preserve flexibility. The simulations were conducted as follows. First, an NVE (N: constant number of particles; V: constant volume; and E: constant energy) ensemble simulation was performed for 1000 steps with a timestep of 0.1 fs. Then, an NVT (T: constant temperature) ensemble calculation was carried out for 1000 steps with a timestep of 0.5 fs. Finally, a timestep of 1 fs was adopted in another NVT simulation for a total of 1.5 ns, where the trajectory was collected every 0.5 ns. The fluid temperature was controlled by using the Nosé–Hoover thermostat algorithm [46,47] and maintained at 300 K.

### 4.3. MDMC Simulations

MD simulations can provide an accurate atom trajectory, which is identified as the ground truth in our MDMC algorithm [32]. The accurate MD trajectories for different atoms were collected, and the trajectory means that the atoms’ spatial points are in the computational domain, so more trajectory is able to represent the spatial distribution of each atom. If enough trajectories are collected, then we are able to determine the probability of atoms touching each spatial point, and we are able to determine the entire probability distribution in terms of spatial coordinates. By discretizing the computational domain, each cell could represent the corresponding spatial state, and the atoms would transfer between these states. The workflow of the MDMC method is schematized in Figure 10. In the MD trajectory, the movement of particles can be obtained. For example, the particle is in spatial state 1 at step 1, and it moves to spatial state 2 at the next step. Thus, the state transition from state 1 to 2 can be recorded. In this way, assuming we have N states in the simulation system, the state transition information between these N states will be collected, generating an N-by-N matrix. After that, this matrix was normalized into a probability transition matrix, with the size as N × N. Then, based on this probability transition matrix produced according to an MD trajectory, the Monte Carlo (MC) simulations could be designed and carried out. During this process, apart from the spatial state reproduction, the residence time of particles can also be counted. Specific details and validations have been discussed in our previous study [32].

## 5. Conclusions

Ion transport through a membrane protein nanochannel was studied by using MD and MDMC methods. The nanochannel was built with graphene layers to facilitate the quantitative analysis, because the graphene model is easy to be constructed and is able to give clear feedback on results, while the realistic protein channel contains a large number of atoms, which requires intensive computational resources. MD simulations were performed to study the ion spatial characteristics during the transport process; the Na ions exhibited density peaks at the nanochannel boundary due to the driving force represented by an external electric field. The two-dimensional potential energy contour of the water was also determined to explain the fluid enrichment phenomenon on both nanochannel sides, showing potential wells and a high potential area caused by atomic collisions. The potential energy contours show that the ions are able to be trapped in the initial part of the channel, which is similar to the ion rejection phenomenon. The MDMC algorithm was based on the MD trajectory, and its predictions agreed well with the MD results on both one- and two-dimensional aspects. Enough MD trajectory data are necessary to obtain a more accurate MDMC result. They have a shorter residence time at a smaller electric gradient, because the ions cannot transport through the channel. Higher electric field gradients facilitate the transport process, and ions are able to finish the transport within a shorter residence time. The residence time reached the highest value at the electric field gradient condition of 0.5 V·Å^−1^. In the simulation, a rare state of ion transport was observed; it was revealed as the specific passing sequence of different components through the protein nanochannel. This dynamic process was described according to the residence time in the nanochannel, which showed that water molecules were trapped in it for the longest time while Cl ion had the shortest residence time. The effect of protein channels is also crucial for waste transport, especially in the case of deficiencies in cellular metabolism. Therefore, the MDMC algorithm can be a more effective and novel tool for a better understanding of ion transport through membrane protein nanochannels, and our results validate its applicability in biomedicine.

## Figures and Tables

**Figure 1 ijms-24-10074-f001:**
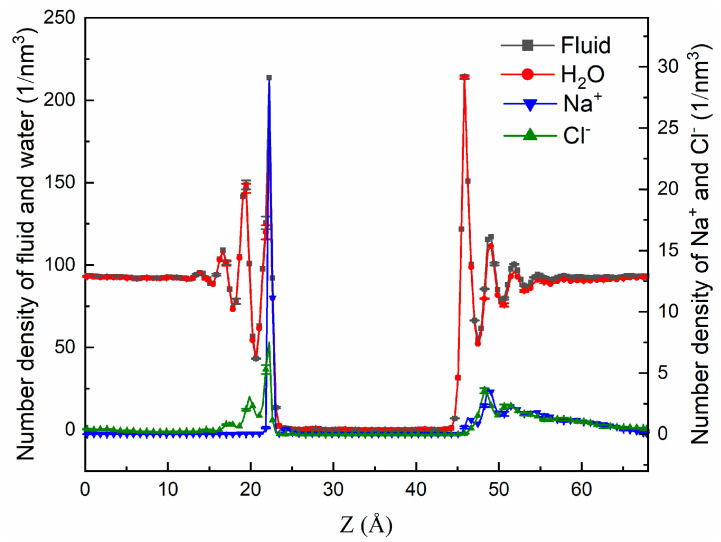
One-dimensional number density profiles of each fluid component.

**Figure 2 ijms-24-10074-f002:**
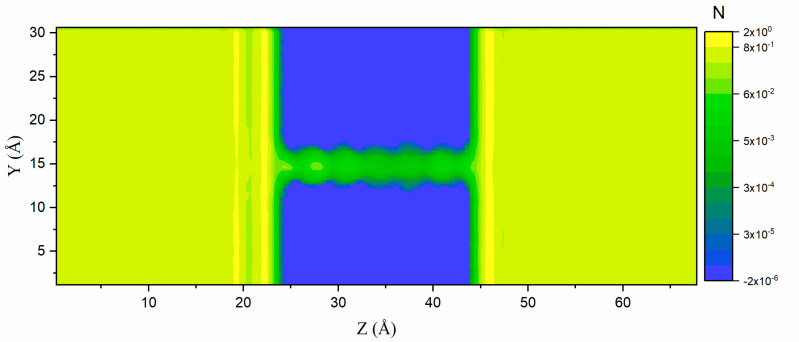
Two-dimensional number density contour of total fluid; here, the atom positions were accumulated and time-averaged. N indicates the time-averaged atomic number.

**Figure 3 ijms-24-10074-f003:**
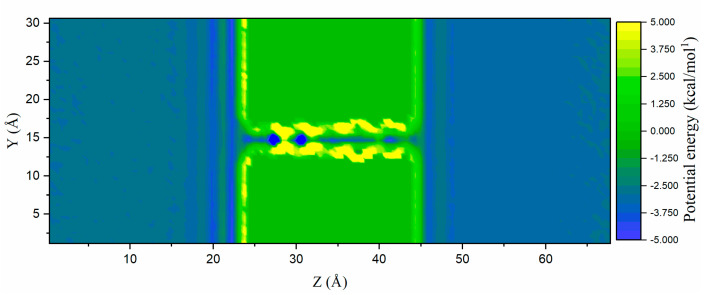
Two-dimensional potential energy contour of the water in the fluid.

**Figure 4 ijms-24-10074-f004:**
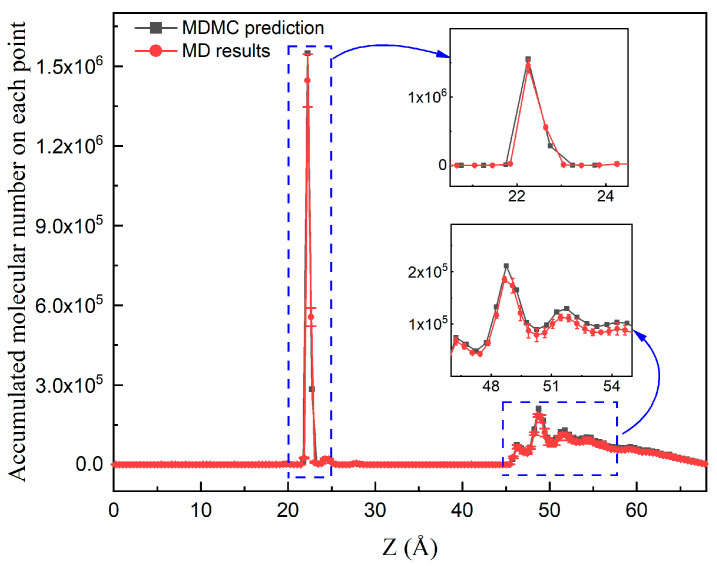
Density profile of the fluid component Na^+^, calculated with molecular dynamics (MD) simulations and predicted with the MD Monte Carlo (MC) algorithm.

**Figure 5 ijms-24-10074-f005:**
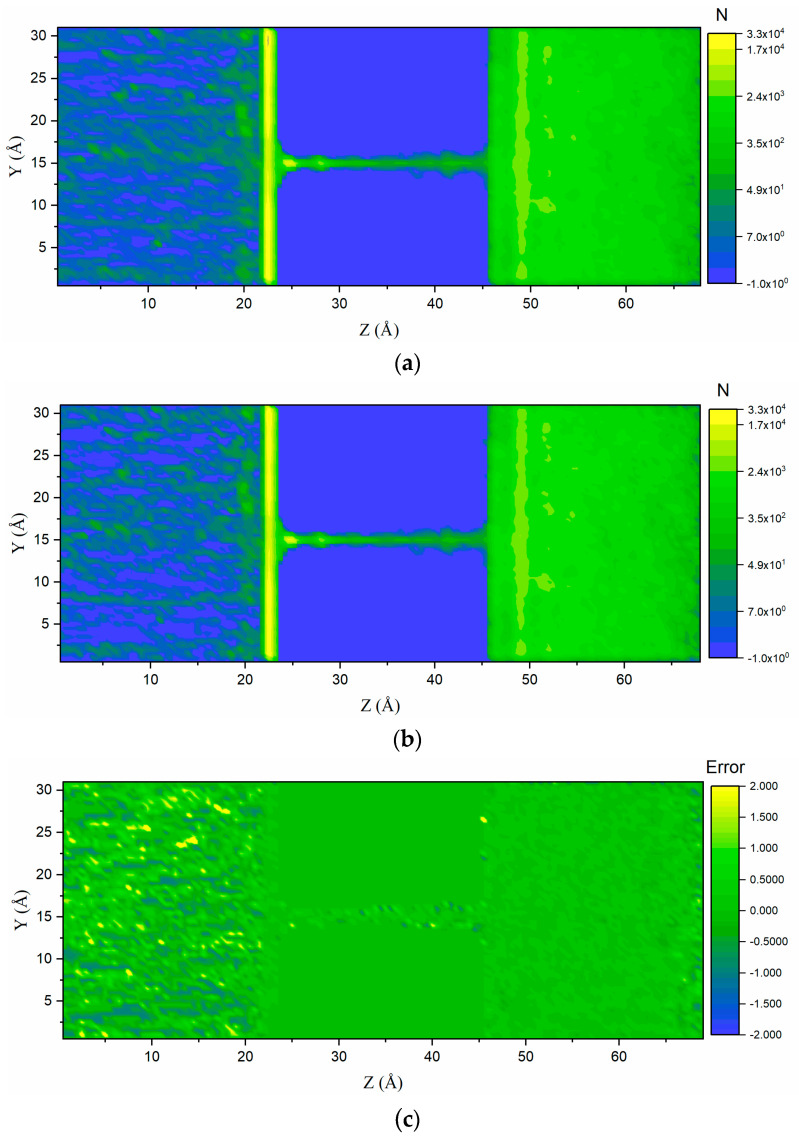
(**a**) Two-dimensional (accumulated) number density contour of the fluid component Na^+^, calculated with molecular dynamics (MD) simulations (**b**) and predicted with the MD Monte Carlo (MDMC) algorithm; N is the accumulated atomic number. (**c**) Error map between the MD and MDMC results.

**Figure 6 ijms-24-10074-f006:**
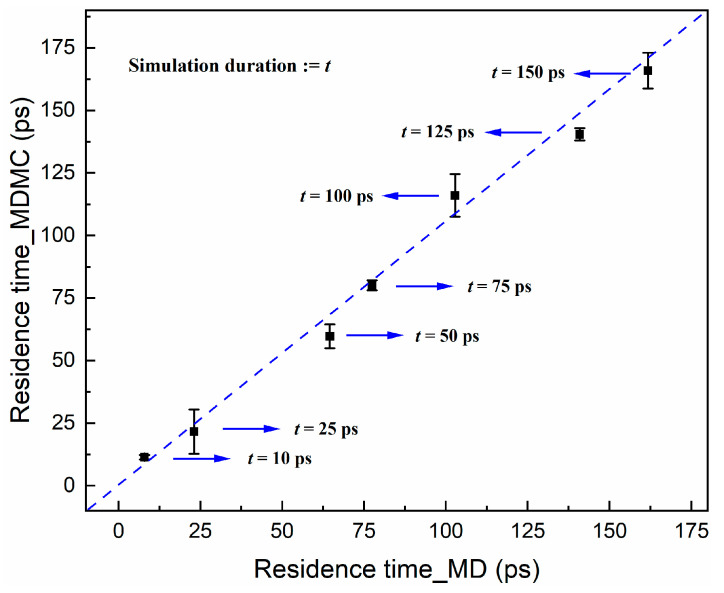
Residence time of Na^+^ in the nanochannel for different simulation durations (MD: molecular dynamics; MDMC: MD Monte Carlo).

**Figure 7 ijms-24-10074-f007:**
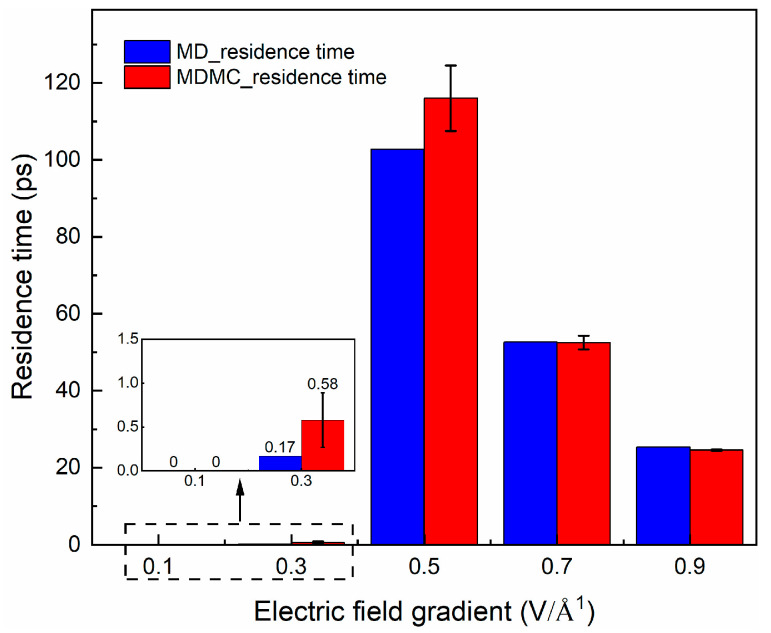
Residence time of Na^+^ in the nanochannel under different electric field gradients (MD: molecular dynamics; MDMC: MD Monte Carlo).

**Figure 8 ijms-24-10074-f008:**
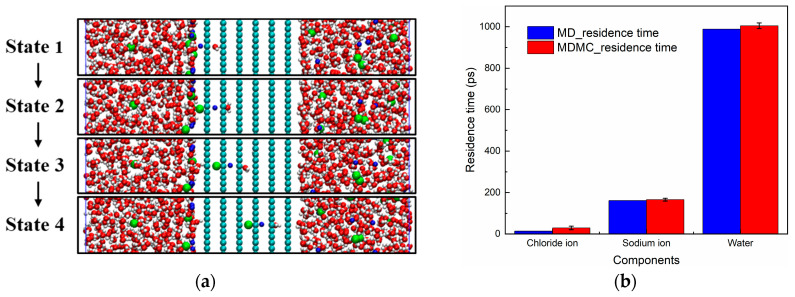
(**a**) Snapshots of different transport states calculated via molecular dynamics (MD) simulations. (**b**) Residence time of different ions in the nanochannel, computed via the MD and MD Monte Carlo (MDMC) methods.

**Figure 9 ijms-24-10074-f009:**
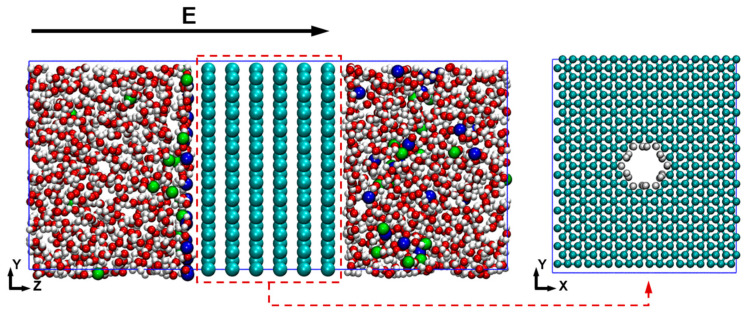
Molecular model of a simplified protein nanochannel in a saline environment, with the following element color scheme: cyan = C; blue = Na^+^; green = Cl^−^; red = O; white = H (E: electric field). The arrow means the range of electrical field and its direction.

**Figure 10 ijms-24-10074-f010:**
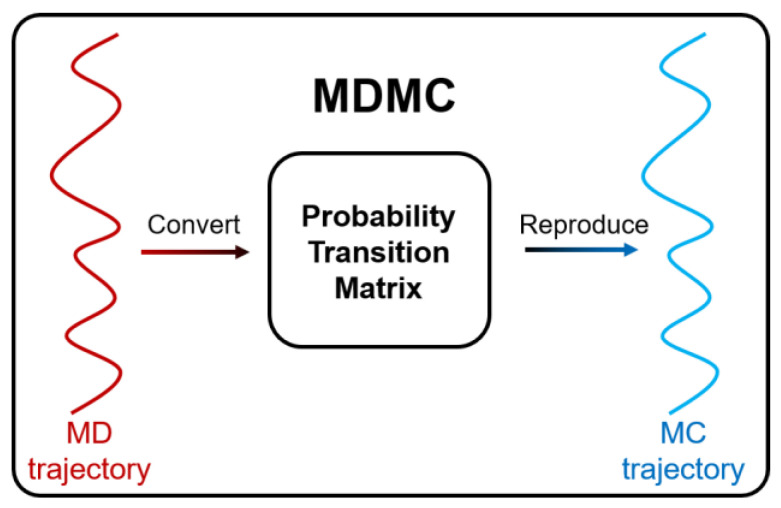
Schematic workflow of the molecular dynamics (MD) Monte Carlo (MC) method.

**Table 1 ijms-24-10074-t001:** Molecular parameters for different molecular components.

Species	Number of Molecules	Molar Mass [g·mol^−1^]	Charge [e]	E [kJ·mol^−1^]	σ [nm]
Na^+^	34	22.99	1	0.738	0.39624
Cl^−^	34	35.453	−1	0.305	0.3915
O (water)	1076	15.9994	−0.82	0.274	0.3608
H (water)	2152	1.00797	0.41	0.013	0.1098
H (graphene)	54	1.00797	0	0.02	0.2995
C (graphene)	1650	12.01115	0	0.064	0.401

## Data Availability

Not applicable.

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
