# Peer review of "Molecular Dynamics Simulations of Ion Transport through Protein Nanochannels in Peritoneal Dialysis"

_ijms, 2023, doi:10.3390/ijms241210074_

Round 1

Reviewer 1 Report

In this manuscript, the authors used MD simulations in combination with a MDMC algorithm to study how ions were transported through membrane proteins in saline environment. There are some important insight from the manuscript. For example, the authors obtained different ion transport states and their residual times, especially some rare states. However, there are some concerns in the design of the MD simulations and the presentation of some results.

Major concerns.

1. The authors used graphene layers to represent protein channels, which seems to be way too simplified. Graphene only contains carbon atoms, while there are other atoms in proteins, especially some strong polar atoms, which can be charged in some cases. Could the authors clarify the advantages of graphene layers except for computational efficiencies?

2. The authors mentioned different transport states in the manuscript several times. Could the authors mention in either Methods or Results how were the states determined? 

3. In Figure 2, the authors converted MD trajectories into PTM. I understand that the authors referenced their previous work on this. But could the authors briefly mention how they did this so that readers would not need to jump from one place to another to read just some general methods?

4. Related to 3, the whole idea of MDMC was similar to Markov State Modeling, where MD trajectories were clustered into different micro states to compute PTM, and then different residual times or transition times between states could be obtained from Markov modeling. Could the authors have some discussions on this?

Minor points.

1. Line 107, there seems to be a missing reference.

Author Response

Reviewer 1:

In this manuscript, the authors used MD simulations in combination with a MDMC algorithm to study how ions were transported through membrane proteins in saline environment. There are some important insight from the manuscript. For example, the authors obtained different ion transport states and their residual times, especially some rare states. However, there are some concerns in the design of the MD simulations and the presentation of some results.

Major concerns.

  1. The authors used graphene layers to represent protein channels, which seems to be way too simplified. Graphene only contains carbon atoms, while there are other atoms in proteins, especially some strong polar atoms, which can be charged in some cases. Could the authors clarify the advantages of graphene layers except for computational efficiencies?

Response: Thanks for pointing this out. Yes, the realistic protein structure and element are problem-dependent. Different polar atoms in proteins are able to induce various effects on the ions transport. Thus, it is better to use a more general model to describe the basic transport process, in order to verify the algorithm’s applicability. The graphene is good at constructing a regular model, suggesting that the quantitative analysis is easy to be addressed, which can provides a clearer comparison. This part of explanation is also enhanced in the manuscript.

  1. The authors mentioned different transport states in the manuscript several times. Could the authors mention in either Methods or Results how were the states determined? 

Response: Thanks for your suggestion. In this study, the transport state is the spatial state, and the entire spatial state distribution can be regarded as the transport state. The spatial state is determined by discretizing the computational domain with a mesh. This is explained in the manuscript.

  1. In Figure 2, the authors converted MD trajectories into PTM. I understand that the authors referenced their previous work on this. But could the authors briefly mention how they did this so that readers would not need to jump from one place to another to read just some general methods?

Response: Thanks for your suggestion. We supply more details relate to the MDMC algorithm in the part of 2.3.  “In the MD trajectory, the movement of particles can be obtained. For example, the par-ticle is in the spatial state 1 at step 1, and it moves to the spatial state 2 at the next step. Thus, the state transition from state 1 to 2 can be recorded. In this way, assuming we have N states in the simulation system, the state transition information between these N states will be collected, generating an N by N matrix. After that, this matrix was normalized into a probability transition matrix with size as N × N.”

  1. Related to 3, the whole idea of MDMC was similar to Markov State Modeling, where MD trajectories were clustered into different micro states to compute PTM, and then different residual times or transition times between states could be obtained from Markov modeling. Could the authors have some discussions on this?

Response: Thanks for your suggestion. Yes, we have supplied more details and discussions regarding this problem.

Minor points.

  1. Line 107, there seems to be a missing reference.

Response: Thanks for your suggestion. It was caused by the super link of the Table 1. We fixed this problem for a better reading experience.

Reviewer 2 Report

Ion channels in cellular life and their relationship with the surrounding environment play an important role in human health.

It is interesting to learn more about their behavior in response to the variation of membrane potentials. In the article "Molecular Dynamics Simulations of Ion Transport through Protein Nanochannels in Peritoneal Dialysis", the authors share their findings with us. But there are some things that can be improved

In introduction, the authors reflect: “Coronavirus disease 2019 patients have also the symptoms of kidney diseases. Thus, more studies about dialysis are necessary and urgent. But they don't really write that relationship or reason for the need.

At least in two lines, there are “Error! Reference source not found”, line: 107, 174

Section Results and Discussion; The authors usually send the reader to different previous or later sections for their explanation, that makes them get lost in the explanation because it takes time to go and return to the reading point. Perhaps they could give an introduction or open a new section once the information is written.

In conclusion, the authors consider the MDMC algorithm as an effective and novel tool to better understand the transport of ions across the membrane, but why? They should relate the importance of channels to deficiencies in cellular metabolism.

Author Response

Reviewer 2:

Ion channels in cellular life and their relationship with the surrounding environment play an important role in human health. It is interesting to learn more about their behavior in response to the variation of membrane potentials. In the article "Molecular Dynamics Simulations of Ion Transport through Protein Nanochannels in Peritoneal Dialysis", the authors share their findings with us. But there are some things that can be improved

  1. In introduction, the authors reflect: “Coronavirus disease 2019 patients have also the symptoms of kidney diseases. Thus, more studies about dialysis are necessary and urgent.” But they don't really write that relationship or reason for the need.

 Response: Thanks for your suggestion. The relevant introduction of the result and motivation is enhanced. Coronavirus disease 2019 patients have also the symptoms of kidney diseases, which means some patients with kidney failure also need the dialysis treatment. Thus, more studies about dialysis are necessary and urgent.

  1. At least in two lines, there are “Error! Reference source not found”, line: 107, 174

 Response: Thanks for your suggestion. It was caused by the super link of the Table 1. We fixed this problem for a better reading experience.

  1. Section Results and Discussion; The authors usually send the reader to different previous or later sections for their explanation, that makes them get lost in the explanation because it takes time to go and return to the reading point. Perhaps they could give an introduction or open a new section once the information is written.

Response: Thanks for your suggestion. We have optimized this problem in the manuscript, to avoid the jump reading.

  1. In conclusion, the authors consider the MDMC algorithm as an effective and novel tool to better understand the transport of ions across the membrane, but why? They should relate the importance of channels to deficiencies in cellular metabolism.

Response: Thanks for your suggestion. “The effect of protein channel is also crucial for the waste transport, especially in the cases of deficiencies in cellular metabolism. Therefore, the MDMC algorithm can be a more effective and novel tool for a better understanding of ion transport through membrane protein nanochannels, and our results validate its applicability in biomedicine.”
